# Induced Effect of Environmental Regulation on Green Innovation: Evidence from the Increasing-Block Pricing Scheme

**DOI:** 10.3390/ijerph18052620

**Published:** 2021-03-05

**Authors:** Zhangsheng Liu, Liuqingqing Yang, Liqin Fan

**Affiliations:** 1College of City Construction, Jiangxi Normal University, Nanchang 330022, China; xiaoliu8033@126.com; 2Economics School, Zhongnan University of Economics and Law, Wuhan 430073, China; 3School of Economics and business, Xinjiang Agricultural University, Urumqi 830052, China; fanliqin1997@126.com

**Keywords:** IBP, energy-efficiency, green innovation, the induced effect

## Abstract

With increasing constraints on resources and the environment, it is of great practical importance to discover and utilize the induced effect of green technology through market-based tools, in order to simultaneously realize economic development and ecological sustainability. Based on unique patent data from 1999 to 2013, this paper examines the induced effect of China’s increasing-block electricity pricing scheme (IBP) on energy-efficient patents and checks whether the effect is neutral or biased. Furthermore, the quality of the induced patents is identified. The results reveal that increased green innovation is strongly related to the IBP scheme. In addition, the induced effect is biased towards green technology such that, apart from autonomous technological advances, the biased effect of IBP induced two more energy-efficient patents per hundred technological patents. However, the quality of the induced innovation is relatively low: compared to high-quality inventions, low-quality utility models showed greater and more significant growth due to the IBP. Our paper provides quantitative insight into the impact of the IBP and indicates that a reasonable pricing scheme can benefit both the environment and the economy.

## 1. Introduction

As high energy consumption and emissions are great challenges for China’s sustainable growth, seeking a way to meet both economic and environmental targets is necessary. Interactions between environmental regulation and innovation have been considered as an important means to overcome resource and environmental restrictions. According to Porter and Linde [1], cleaner and energy-efficient technologies are induced by strict environmental regulation, in order to offset compliance costs. Such technology is called green innovation [2], which reduces the consumption of energy and raw materials, reduces emissions, and improves recycling. Since then, studies on green innovation induced by environmental regulations have emerged. Lanoie [3] and Yang et al. [4] have assessed the empirical validity of the Porter Hypothesis and found strong support for environmental policies triggering innovation, as reflected by research and development (R&D) expenditures.

Meanwhile, advances in green technology may vary from each other: They can be neutral when technological innovation reduces the factors (resource, labor) used for one product/service in the same proportion, or they can be biased; that is, the amount of a particular factor for one product/service drops in a larger proportion than other factors, due to technological innovation [5,6]. As innovations are made based on existing technology, the former is always biased towards the latter, which is called path dependence. Acemoglu et al. [7] introduced endogenous and biased technical change in a growth model with environmental constraints, and found that the optimal policy, which can redirect innovation in a cleaner way, involves both carbon taxation and research subsidies.

As an increased quantity of green technology does not necessarily lead to true innovation, scholars have shown interest in the quality of innovation, by which we mean to what extent a patent makes breakthroughs and improvements, instead of simply changing appearance over existing models. Aghion et al. [8] discovered the reciprocal causation between environmental regulation and quality of innovation, as measured by types of patents. Hu et al. [9] studied the impact of China’s carbon emissions trading system (CETS) on the quantity and quality of innovation and showed that the CETS has mainly promoted innovation quantity and low-quality innovation. Consequently, we hope to shed light on three questions when evaluating the effectiveness of a particular environmental regulation, with regard to green technology: Does the policy induce green innovation? Is the induced effect neutral or biased? Additionally, what is the quality of the induced green innovation?

Since the “Eleventh Five-Year Plan” period, in order to control the energy consumption and pollution emissions that accompany economic growth, the Chinese government has strengthened its policy constraints and gradually formed a policy system for energy conservation and emissions reduction. In this system, market-based policy instruments have been placed with high expectations. In this situation, the increasing-block electricity pricing (IBP) scheme for residential electricity consumption has been experimented with in Zhejiang province in 2004, as well as in Fujian and Sichuan provinces in 2006, and implemented nationwide in 2012. The pricing baselines vary among provinces (Table 1), but they follow some common rules. The scheme is set with three levels: The floor level covers electricity usage for basic electrical appliances and charges for the cost of generation and transmission, the middle level covers over 95% of residential consumption and charges an extra 10% premium, and the top level is applied to wealthy families and charges 150–200% of the middle level price. Generally, unit electricity price is increased for most families.

Related studies have mainly focused on the demand side, including consumer behavior [10] and awareness of environment protection and policy efficiency [11,12]. Generally, the expected targets have been reached [13]. Nevertheless, these studies have only considered the direct impact of IBP on energy price and demand, while forgetting that the impact offers great incentives to innovate for less energy consumption and higher efficiency; that is, the IBP may indirectly induce green innovations. Therefore, it is of great practical importance to study the induced effect and assess the effectiveness of IBP.

Undoubtedly, the above studies have provided comprehensive insight into the induced effect and effectiveness of the IBP scheme. However, the induced effects of administrative regulation and economic policies, such as tax benefits, subsidies, and emissions trading systems, have already been discussed, while that of the IBP scheme has been ignored, which significantly changes the price of electricity. Furthermore, existing studies have focused on biased technological advances induced by energy price, demand, and path dependence, instead of the IBP. Even the few studies focused on the IBP have implicitly assumed that the marginal rate of substitution does not change at all. However, the possibility that the biased induced effect exists under the IBP scheme has not been taken into consideration. In addition, studies involving both the quantity and quality of green innovations are seldom seen.

Using patent data from 1999 to 2013, this paper explores the effects of the IBP scheme on green innovation, in terms of quantity and quality, based on the estimation of generalized differences–in-differences (DID). First, we confirmed that the IBP induced green innovation (induced effect). Then, we checked whether the induced effect was neutral or biased; that is, whether the energy-efficiency patents were increasing at the same rate as other kinds of patents, or at a faster speed. Finally, the quality of induced innovations was studied, in terms of types of patents.

This paper differs from other studies in three ways: Firstly, we not only examine the relationship between booming green innovation and the IBP scheme, but also provide mechanisms to explain the induced effect from three perspectives. We further distinguish the biased advancement towards green technology from the autonomous development of technology. To isolate the biased effect of the IBP, we test energy-efficient patent count per hundred technological patents. Some previous studies [14,15] have empirically tested how the energy price induces technological changes, while the rest have evaluated the effectiveness of China’s IBP scheme with respect to demand and environmental awareness, and barely any attention has been paid to the relationship between the IBP and green innovation. Even those focused on the IBP have implicitly assumed a constant marginal rate of substitution of factors, blind to the possible biased effect. This study aims to address these gaps.

Secondly, this paper goes beyond identifying the quantity of green patents and further looks into innovative quality by differentiating patent types between high-quality inventions and low-quality utility models. Although scholars have gradually become aware of the impact of environmental regulation on innovation, and further classified technological innovation as green innovation and non-green innovation [16], or process innovation and product innovation [17], they have not yet taken the difference in innovation quality into consideration, due to the complex characteristics of the technology. Moreover, some studies have used patent citations to measure the quality of innovation in China [18], but this measurement is doubtful, as the China National Intellectual Property Administration (CNIPA) only provides details of patents from 1985 to 2012 [19], and the sample time span is not long enough for this paper. Green innovation performance—which, in essence, is productivity with pollution as undesirable output—has also been used as an indicator for innovation quality [20,21]; however, it is not easy to make any conclusions using such measurements, as current standards for both high- and low-quality innovation are ambiguous.

Finally, this paper employs the generalized DID for baseline regression [22] and conducts pre-existing trend tests, in order to ensure the common trend assumption. Generalized DID is employed specifically to identify the effects of policies implemented at different times in different areas [23,24,25]. The conventional models for induced effect analysis are ordinary least squares (OLS) and DID regression [2,26], which always lead to omitted variables and endogeneity bias. For robust conclusions, this paper identifies a key determinant of IBP selection and checks its effect on the balance between the treatment and control groups. Then, we employ the counterfactual method, which is useful in policy analysis. A narrow time window is also used to support the conclusion. In addition, we form our unique dataset by referring to the International Patent Classification (IPC) Green Inventory and making a specific selection of topics related to the IBP. This dataset enables us to analyze induced innovation from the perspective of supply, an area which has seldom been set foot in by the existing literature, and frees us from data limitations. 

The remainder of this paper is organized as follows: Section 2 reviews the related literature, China’s IBP scheme, and the mechanisms of the induced effect. Section 3 provides the data and method, including the data source, variable explanation, and identification strategy. Section 4 presents the basic empirical results and robustness test. The discussion is in Section 5 and the conclusion is given in Section 6.

## 2. Literature Review and Background

### 2.1. Induced Effect

It is often said that regulation is harmful to innovation. This is true, in the limited sense that regulations raise the costs of production [26,27] and of compliance [10], and that such expenditure and resources might be used instead to foster innovations. It is also true in the sense of curtailing freedom to innovate and crowding out innovation [28].

However, properly designed environmental standards can enhance innovations that partially (or more than fully) offset the costs of complying with them [29]. As a profit-motivated activity, R&D activities can respond to changing cost in the way that inventions related to a particular factor (e.g., electricity or energy) are spurred, in order to economize the use of that factor, which has become relatively expensive [6,30]; in particular, energy price. Such technological change that might be introduced in the face of environmental policy is called an induced effect [15]. Johnstone et al. [31] found that public policy plays a significant role in determining patent applications and that different types of policy instruments are effective for different renewable energy sources. The increased stringency of environmental regulation spurs increased innovative activities by firms [6].

The IBP scheme has been excluded by previous works on the induced effect, despite changes in the price of electricity as a result of its existence. In this work, we analyze the scheme and overcome this deficiency.

### 2.2. Biased Induced Effect

When the marginal rate of substitution between factors changes, the technological change is biased; that is, innovation related to certain factors is growing at an increasing rate, in addition to the autonomous technology improvement [32]. Biased innovation can be trigged by factor price [33] and increased demand, as well as by path dependence [7]. Path dependence indicates that the stock of existing experience and technology decides and makes great contributions to innovation, such that the technological advances are standing on the shoulders of giants. Such ideas have been supported by studies on energy prices and environmental policy instruments, such as tax concessions and subsidies [8,34]. For example, Tang et al. [35] claimed that only when the rate of carbon tax and carbon reduction subsidy reaches a certain extent, will individuals (or producers) redirect their technical change towards “clean” energy innovation. Greak and Heggeda [36] stated that subsidies to R&D are not as effective as carbon taxes, under the assumption of long validity.

### 2.3. Quality of Green Innovation

With the deepening of research, scholars have investigated the quality of innovation more carefully. Nesta et al. [37] concluded that environmental regulation plays a crucial role in inducing high-quality green patents. Hu et al. [9] found that the CETS only promoted the innovation quantity and high-quality innovation of state-owned share firms, large-size firms, and eastern-section firms. Guo et al. [38] argued that environmental regulation has a significant effect on both invention and utility models but is a little bit more robust on the impact on green process innovation. Although the number of patent-based innovation in China has exploded, there still exists a big gap, in terms of the innovation quality, compared to developed countries, such as Europe and America [39].

#### 2.3.1. IBP Scheme in China

Existing research on the IBP has focused on demand and policy efficiency [12,13]. The IBP increases the price elasticity of users [10] and provides a good incentive to reduce the use of electricity by families [13,40]. Residents began to pay attention to their own electricity consumption behavior and their awareness of energy saving was effectively awakened [40]. Generally, the expected targets have been reached [13]. However, these works have implicitly assumed that electricity consumption changes with usage, ignoring advances in energy-efficient technology. In other words, a biased induced effect possibly exists under the IBP scheme. Therefore, it is of great practical importance to study the induced effect and assess the effectiveness of IBP.

#### 2.3.2. Mechanism of Induced Effect

The induced effect of IBP is similar to an exogenous price added to the energy price. Figure 1 analyzes how the IBP scheme induces energy- and power-efficient innovation from three perspectives. The pre-condition is that, with more electricity consumption, the electricity payments of residents increase, which, in turn, makes consumers more willing to pay for energy-efficient products. From the micro-perspective, the IBP raises the price of electricity, thereby improving the market recognition of energy-efficient products and, thus, stimulating enterprises to produce energy-saving products. From a macro-perspective, the IBP scheme needs to be effectively transmitted to enterprises and significantly enhance the market demand for energy-efficient products to form economic incentives, thus eventually inducing energy-efficient R&D activities.

## 3. Data and Method

### 3.1. Data

The definition, source, and measurement of all covariates are shown in Appendix A (All data is made available in the excel file named data for replications.).

#### 3.1.1. Patent Data

Researchers have not yet reached an agreement on the definition of green innovation [41]. Green total factor productivity [42] and technological patents are common indicators of green innovation [43], and the latter is employed in this paper, due to its straightforwardness and accuracy. Energy-efficient patents are specifically isolated in this analysis, as they directly reflect the impact of the IBP.

Patent data was made available by the State Intellectual Property Office (SIPO) and includes patent title, granted number, applicant name, patent type, and International Patent Classification (IPO) code. For every patent, one or more IPC codes was assigned to classify its function. Thus, we used the IPC codes to select energy-efficiency patents from all technological patents. To identify the energy-efficient IPC code, we referred to the IPC Green Inventory, and the specific selection of topics is provided in Appendix A. We narrowed the definition of green innovation down to reducing the consumption of energy and energy storage, which are possible consequences of the IBP.

An important issue about Chinese patent filing is the insufficiency and unavailability of citation information, which is generally used as an indicator of patent quality. Some studies have addressed this issue with datasets filed jointly at the SIPO, the United State Patent and Trademark Office (USPTO), and the European Patent Office (EPO) [21]; however, such patent families are extremely rare and take a much longer time from application to publication than those at the SIPO. According to the Patent Law of the People’s Republic of China (PRC), patents are divided into three categories: Utility models, inventions, and outline designs. Inventions involve the highest level of innovation and original creation. Invention applications are the most rigorous and time-consuming and, so, the patent office only files patents with big breakthroughs and remarkable improvements over existing models. Utility models are new models that make minor improvements on the shape, structure, and utility of existing inventions, which can be quickly produced and applied. Outline designs are rarely employed in the field of energy efficiency [9]. For patent quality measurement, we differentiated patent types between inventions (high-quality) and utility models (low-quality). Allowing for 1–3 years for patents to be granted, patent data were searched for in October 2016, in order to obtain a full sample of patents during the observation period of 1999–2013.

#### 3.1.2. Control Variables

Economic variables: The economic variables included the level of the economy, the structure of the economy, the economic environment, and international trade, which were, respectively measured as the natural logarithm of the total Gross Domestic Product (GDP) (in constant 1998 prices), the ratio of secondary industry to service sector, proportion of scaled firms at a loss in all firms, and ratio of foreign direct investment (FDI) to total GDP. The data were retrieved from the *Easy Professional Superior database* (EPS)database and the *China Statistical Yearbook*.

Technological variables: The technological variables included human resources and investment in R&D, which were, respectively measured as the average education years of citizens over the age of 6 and the natural logarithm of the internal expenditures on R&D (in constant 1998 prices). The data were retrieved from the EPS database.

### 3.2. Identification Strategy

#### 3.2.1. Identifying Assumptions and Checks

The key challenge in identifying the effect of the IBP scheme is selecting appropriate control groups for the treatment group. The validity of the generalized DID and the causal interpretation of the results rely on two assumptions: (1) There is a common pre-existing trend in patents between experimental provinces and non-experimented provinces, and (2) the cross-province IBP adoption is randomly selected.

Following Beck et al. [44] and Song et al. [45], we used a series of dummy variables to test the pre-existing trend hypothesis:(1)$$patent_{it}=\beta_{0}+\beta_{1}B_{it}^{3}+\beta_{2}B_{it}^{2}+\beta_{3}B_{it}^{1}+\beta_{4}Current_{it}+\beta_{5}A_{it}^{1}+\cdots +\beta_{10}A_{it}^{6}+u_{i}+\mu_{t}+\varepsilon_{it},$$
where $patent_{it}$ indicates the technological patent/energy-efficient patent count of province *i* in year *t.*$\;B_{it}^{j}$, $Current_{it}$, and $A_{it}^{j}$ are dummy variables, where $B_{it}^{j}$ equals 1 when year *t* is the j^th^ year before province *i* implements the IBP scheme,$\;Current_{it}$ equals 1 when year *t* is the year that province *i* implements the IBP scheme, and $A_{it}^{j}$ equals 1 when year *t* is the j^th^ year after province *i* implements the IBP scheme.$\;u_{i}$ represents the province fixed effects, $\mu_{t}$ is the time fixed effects, and $\varepsilon_{it}$ is the error term. If the trends between the treatment and control groups before treatment are insignificantly different, then the common trend hypothesis stands. 

Appendix A plots the implementation of the IBP over time and the 95% confidence intervals (CI), adjusted for province-level clustering. The estimated coefficients from the regression of Equation (1) are shown as dots in the middle of lines, which suggest that both energy-efficient patents and their proportion in all technological patents showed no obvious variations before the IBP, but increased dramatically since its introduction between the treatment and control groups. We also plot the common trend (Appendix A) and checked the average growth rates of both treatment and control groups before and after the IBP (Appendix A).

However, our major concern was that the considered provinces were not randomly selected. Thus, the divergence in Appendix A after the IBP may have been caused by some pre-existing differences between the considered and non-considered provinces. To address this identification challenge, residential electricity consumption was identified as a key determinant in the selection of IBP provinces. Then, differential trends in outcomes between the experiment provinces and non-IBP provinces after the adoption of the IBP caused by that determinant were controlled for. We followed Agarwal and Qian [46] and checked the balancing to verify whether controlling for this variable led to a better balance between the treatment and control groups. Appendix A shows the key selection criteria. On average, residents in the considered provinces consumed more electricity than residents in non-IBP provinces, illustrating that resident electricity consumption plays an important role in determining the treatment status. Then, comparisons between the treatment and control groups, in terms of control variables in the initial year, showed that there were significant differences between the considered provinces and non-IBP provinces in both economic and technological dimensions. On average, the economy in considered provinces was more developed and optimized. IBP provinces had a better economic environment, developed international trade, and more investment in R&D, but citizens in the IBP provinces were less educated. However, as shown in Appendix A, after controlling for residential electricity consumption, none of these variables exhibited any statistically or economically significant differences between the treatment and control groups. The treatment and control samples were balanced, which is important for the causal identification.

Beyond that, we restricted the time window to 5 years, increasing confidence in the comparability of the treatment and control groups. We also conducted a counterfactual test by changing the treatment group and control group, as well as the start and end points.

There were also some limitations when using the method, as it was a great challenge to create appropriate control groups. If the control group is not a valid counterfactual for what would have happened to treatment group in the absence of the treatment, the results will be biased.

#### 3.2.2. Estimation of the Induced Effect on Green Innovation

We used the following specifications to confirm the causal relationship between the IBP and increasing energy-efficient patent:(2)$$patent_{it}=\alpha_{0}+\alpha_{1}policy_{it}+\alpha_{2}X_{it}+u_{i}+\mu_{t}+\varepsilon_{it},$$
where $patent_{it}$ indicates the technological patent/energy-efficient patent count of province *i* in year *t*, and $policy_{it}$ is the experiment dummy, which is set to 1 at year *t* if province *i* starts the reform in year *t* and 0 before year *t*. The term$\;policy_{it}\;$captures the effect on energy-efficient patents in the experimental provinces during the period 1999–2013 and, so, $\alpha_{1}$ is the parameter of primary interest in this section. A set of control variables that affect patents, such as industry structure, are adopted as $X_{it}$. $u_{i}$ is the province fixed effects, $\mu_{t}$ is the time fixed effects, and $\varepsilon_{it}$ is the error term.

#### 3.2.3. Estimation of The Biased Effect towards Green Innovation

Green innovation is a part of technological development, which is expected to grow autonomously with technological advance at a fixed ratio of all technological patents, even in the absence of IBP. Therefore, we utilized the following specifications to confirm the causal relationship between the IBP and increasing ratio of energy-efficient patents: (3)$$Patent_{it}=\gamma_{0}+\gamma_{1}policy_{it}+\gamma_{2}X_{it}+u_{i}+\mu_{t}+\varepsilon_{it},$$
where $Patent_{it}$ indicates the energy-efficient patent count per hundred technological patents in province *i* in year *t*, and $policy_{it}$ is the experimental dummy, which is set to 1 at year *t* if province *i* starts the reform in year *t* and 0 before year *t*. The term$\;policy_{it}\;$captures the effect on energy-efficient patents in the experimental provinces during the period 1999–2013, such that $\gamma_{1}$ is the parameter of primary interest in this section. 

The counterfactual specifications in this section are as follows:(4)$$Patent_{it}=\phi_{0}+\phi_{1}policy_{it}+\phi_{2}X_{it}+u_{i}+\mu_{t}+\varepsilon_{it},$$
where $patent_{it}$ indicates the number of energy-efficient patents per hundred technological patents in province *i* in year *t.* The term$\;policy_{it}\;$captures the effect of the cancellation of IBP on energy-efficient patents, such that $\phi_{1}\;$is the parameter of primary interest for the robustness test.

#### 3.2.4. Estimation of the Effect on Quality of Green Innovation

Along with the quantity of green innovation, the depth of green innovation is an essential factor to evaluate the induced effect of the IBP. Utility models and inventions are frequently concerned in the context of the IBP. Specifically, more advanced, original technology is involved in inventions, while only minor modifications are applied to a utility model. Therefore, we conducted comparisons between utility models and inventions, in order to identify the extent to which green innovation has been motivated by the IBP. The baseline specification was as follows:(5)$$Dpatent_{it}=\lambda_{0}+\lambda_{1}policy_{it}+\lambda_{2}X_{it}+u_{i}+\mu_{t}+\varepsilon_{it},$$
where $Dpatent_{it}$ indicates the number of utility models/inventions in 100 energy-efficient patents in province *i* in year *t*. $policy_{it}$ captures the induced effect on utility models/inventions of energy-efficient patents in the experimental provinces during the period 1999–2013, such that $\varphi_{1}$ is the parameter of primary interest in this section.

## 4. Results

### 4.1. The Induced Effect of IBP on Green Innovation

The induced effects on green innovation are shown in Table 2 and Figure 2a. The results show that IBP had a significantly positive effect on all technological patents and energy-efficient patents of the experimental provinces at a 1% level. Specifically, all technological patents and energy-efficient patents were increased by 30% and 58%, respectively, due to the IBP, indicating that the induced effects were much more significant on green innovation than other patents.

As expected, the effects of total GDP, the ratio of secondary industry to service sector, and expenditures on R&D were significantly positive for all technological patents and energy-efficient patents. The ratio of FDI to total GDP was negatively associated with green innovation; that is, FDI tends to flow to countries with less energy-efficient patents and certainly less environmental constraints, which implicitly supports the Pollution Haven Hypothesis (The Pollution Haven Hypothesis describes the phenomena that pollution-intensive firms tend to operate and engage in production in areas or countries with lenient environmental standards.). Notably, the proportion of scaled firms at a loss in all firms was negatively related to green innovation count, while showing no influence on all technological patents. The explanation for this may be that more scaled firms at a loss reflects a declining economy and employment and, in the setting of the IBP scheme, households would rather reduce their total utility of electricity in order to remain at the floor level where the unit price is unchanged. Consequently, the consumption of new energy-efficient models has dropped, failing to motivate green innovations. The effect of education was not significant, as it takes years for education to impact green innovations. In addition, China’s professional education system needs improvement.

We also tested our conclusions by repeating the process in the narrow time window from 2003 to 2007, during which only 3 provinces were used, and the results were supportive of those detailed above (columns (1) and (2) in Appendix A).

### 4.2. The Biased Effect towards Green Innovation

Even in the absence of IBP, the total energy-efficient patent increased with continuous inputs, as a result of the technological development that is described as autonomous growth, and the former estimates may have mistakenly captured the effects of common technology development, rather than that of the IBP. Therefore, the proportion of green innovation in all technological patents was used to isolate the biased effect from autonomous growth.

Columns (1) and (2) in Table 3 and Figure 2b show the biased effect of the IBP that the causal relationships between the IBP and the increased ratio of energy-efficient patents to all technological patents was significant at the 1% level. The energy-efficient patent count per hundred patents in the experimental provinces was 1.98 higher than that in otherwise identical provinces, strongly proving that the IBP induces energy-efficient patents to grow at an increasing rate, which provides evidence for the biased effect toward green innovations. For the control variables, the share of FDI in total GDP and the ratio of firms at a loss had negative effects on green innovation, consistent with the discussion in the previous section.

To avoid omitted variables and bias, the counterfactual method was employed for the robustness tests. The basic idea is to exchange the start and end points of the experimental period, as well as the control group and the treatment group. In this case, 2013 was set as the start point and 2006 the end point. In 2012 (the year the IBP was widely adopted in China), IBP was assumed to be abolished in the nation (treatment group), except in Sichuan, Zhejiang, and Fujian provinces (control group). Such assumptions resulted in a 1.16-fold reduction in energy-efficient patents (column (4) in Appendix A), providing strong evidence for the reliability of our conclusions.

We also tested the bias effect by narrowing the time period from 15 years to 5 years (2003–2007), and the results were supportive (column (3) in Appendix A).

### 4.3. Quality of Induced Innovation

Table 3 support the causal effect of the IBP on only utility models at the 5% level. Columns (3) and (4) in Table 3 and Figure 2c suggest that, due to IBP, the ratio of utility models to all energy-efficient patents significantly increased; however, that of inventions did not appear to be associated with IBP. Despite different significance levels, the coefficient of utility models was larger than that of inventions. In other words, the induced effect predominately resulted not in advanced new models, but in minor improvements to existing models. There may be two reasons for this: First, the increased price of electricity is not high enough to motivate firms to invest in energy-efficient inventions, which are far more costly than utility models. Second, the IBP has not been implemented long enough for the new ideas to transfer to inventions. Instead, to advertise and push up sales, utility models can be applied to products quickly and, consequently, have become mainstream. 

## 5. Discussion

Compared with existing studies, this paper is unique, in terms of its objectives, method, and results. In terms of objectives, the induced effect of energy price has gained much attention from many scholars. Most of the related studies have focused on innovation induced by an increase in energy price [29] or carbon taxes [47,48], while few have highlighted the shocks due to changing electricity prices—such as those induced by China’s IBP scheme, which has brought strong incentives to energy-related areas. Among those studies considering the IBP scheme, most have examined the effect on demand for electricity and the efficiency of the scheme [49]; however, the induced innovation and biased induced effect have rarely been included in the scope of these studies. To fill this gap, the objective of this paper was to analyze the effects of the IBP scheme on green innovation. Energy-efficient patents, which are used herein as a proxy for green innovation, have received considerable attention, as they are expected to respond dramatically to changes in electricity prices. The biased induced effect is isolated from autonomous technical changes. Quality analysis of patent category has seldom been included in studies on the induced effect, while we discussed the favorability of the IBP for utility model and invention energy-efficient patents separately. We also considered the reality of booming energy-efficient patents, providing new practical insights for the induced effect.

In terms of the method, we employed generalized DID [50] for the baseline analysis, while the counterfactual method and a narrow time period were used for the robustness tests. Generalized DID is typically employed to identify the effects of policies implemented at different times in different areas [3,4,10], treating these policies as quasi-natural experiments. The conventional models for induced effect analysis are OLS and DID [2,27] which always lead to the omission of related variables and endogeneity bias. To make up for such shortcomings, the generalized DID method was adopted, in order to explore the induced effects of the IBP scheme on green innovation, focusing on the policy, which was experimented step-by-step in certain areas. The pre-existing trend and the key determinants of IBP selection were tested, in order to support the assumptions of the identification strategy. For robust conclusions, the counterfactual method was employed, by switching the start point of the experiment period with the end point and exchanging the control group with the treatment group. The results were also replicated using a narrowed time window of 5 years. In terms of data, the response of the supply side to technological innovation has been rarely discussed in the existing literature, as data on the demand side (consumers) are easily obtained, while data on the supply side (electricity providers) regarding the influence on innovation are not readily available [26]. By matching the topics chosen from the IPC Green Inventory and the IPC code offered by SIPO, we constructed a plausible measurement of green innovation and, so, the results we obtained with this combination of methods have higher practical value.

In terms of the results, we found that China’s IBP scheme indirectly promoted green innovation. Some scholars have reported similar findings [26,51]; for example, the U.S. energy price has strongly significant positive effects on energy-efficient innovation. Jaffe et al. [6] and Newell et al. [52] reported that a significant amount of innovation has been due to changes in the energy price and, when the real energy price is falling, air conditioners become less energy efficient. However, some studies have shown different results [26,53]. Hahn and Stavins [54] claimed that the Best Available Control Technology (BACT), targeted at higher technology standards, might freeze the development of technologies, as this regulation provided no financial incentive for businesses to exceed the control target and the adoption of new technologies was discouraged. One possible explanation for the deviation from our results is that the BACT regulation is a “command-and-control” regulation, which tends to force firms to take on a similar share of the pollution/energy-efficient control burden, regardless of the cost. In contrast, the IBP is a market instrument which allows for relatively abundant flexibility, in terms of the means used to achieve energy efficiency. The results of this paper also indicate that the IBP scheme has increased the proportion of green innovation, where the induced effect is biased towards green innovation. This result is in line with other studies related to the induced effect: Popp [15] suggested that the efforts of doubling U.S. government energy R&D spending led to an increased energy patent count, which crowded out other types of patents. This paper provides evidence supporting related studies [39,55], which have stated that the IBP scheme had a more profound influence on low innovative utility models than on highly innovative inventions, considering either the significance level or the size of the coefficient; revealing that, in spite of rapid growth, green innovation primarily appears in the form of minor improvements to previous models, rather than completely new innovations.

## 6. Conclusions

This paper fills a critical gap in the existing literature on the link between the IBP scheme and green innovation for the first time. In particular, we examined the induced effect of the IBP on green innovation, through the use of the generalized DID method, and offered explanations. Furthermore, we checked the biased effect of the IBP in order to isolate the effect of IBP from autonomous technological development. Afterwards, the quality of induced innovation was identified by category. Our main findings are as follows: (1) The induced effect was significantly positive on both technological patents and energy-efficient patents. (2) Technological advances were biased towards green innovation, such that, apart from autonomous technological advances, the biased effect of IBP induced two more energy-efficient patents per hundred technological patents. (3) The quality of induced innovation was relatively low, where low-quality utility models showed greater and more significant growth due to IBP, compared to high-quality inventions.

For historical reasons, gentle effort has been put into the reform of the energy sector. Our findings offer practical policy implications, in response to such slow progress. Firstly, from the demand side, even a subtle growth of electricity price yields a significant effect on green innovations. Additional similar pricing schemes may be applied to residential water or natural gas sectors, which may not only help to save energy but can also contribute to green technology innovation.

Secondly, from the supply side, for a long time, energy price distortion has been one of the reasons why China’s petrochemical energy consumption cannot be efficiently reduced. Speeding up the establishment and improvement of market-oriented management systems in the energy sector is not only a necessary step in energy management, but also an important guarantee to promote technological innovation.

Thirdly, China has made great progress in technological innovation; however, most of these innovations are low-quality utility models. Therefore, it is of great significance to optimize the incentive mechanism for technological innovation and to induce more high-quality innovations. 

Under the context of the IBP scheme, the conclusion and implications are only applicable to China, at present. Future works should investigate the generalization of these findings in China to the period after 2012 and in other developing countries. For now, the induced effect predominately impacts utility models, while true innovation has played a minor role. It is of practical value to explore whether profound technical changes have occurred in energy-related fields since the wider adoption of the IBP in China. As has been proven through cases in developed countries, non-linear pricing is useful for the conservation of primary energy, apart from secondary energy.

## Figures and Tables

**Figure 1 ijerph-18-02620-f001:**
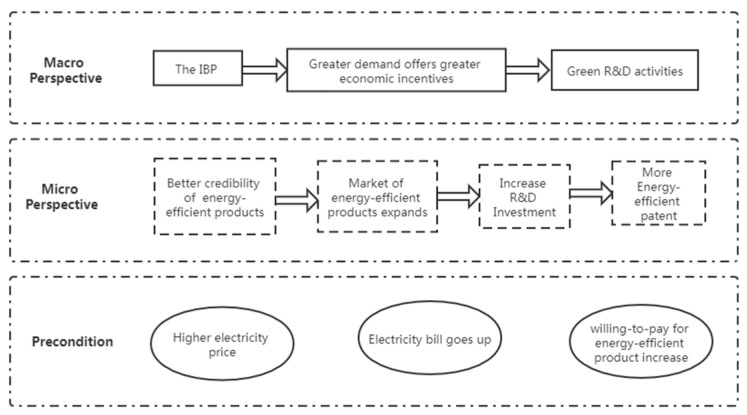
Mechanism of the induced effect of the IBP.

**Figure 2 ijerph-18-02620-f002:**
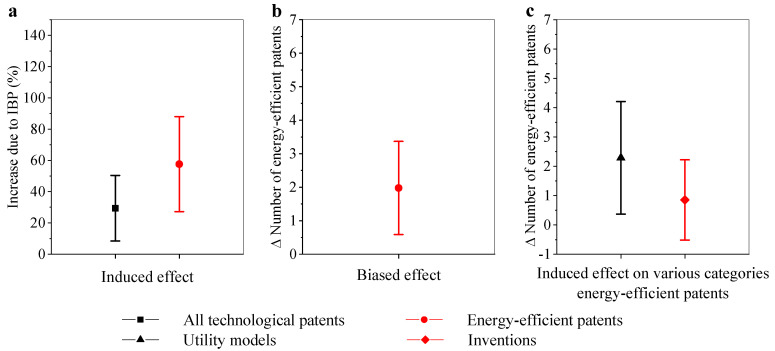
The effect of IBP on patents: (**a**) The induced effect of IBP on the number of technological patents/energy-efficient patents and 95% confidence intervals, (**b**) the biased effect on the number of energy-efficient patents per 100 technological patents and 95% confidence intervals, and (**c**) the induced effect on number of utility models/inventions per 100 energy-efficient patents and 95% confidence intervals. All results in the figure used the method of generalized DID with controls.

**Table 1 ijerph-18-02620-t001:** Pricing rules in the experiment provinces and the remaining provinces.

	Level	Price (yuan/kW·h)	Consumption (per Month/Family)	Coverage (%family)/Average Consumption
Zhejiang Province in 2004	Floor level	Baseline price	Under 50 kW·h	39.49%
	Middle level	Baseline price + 0.03	51–200 kW·h	26.93%
	Top level	Baseline price + 0.1	Over 200 kW·h	33.58%
	Note: The electricity price under the IBP in Zhejiang Province was predicted to increase by 0.032 yuan/kW·h. IBP is not applicable to schools.			
Fujian Province in 2004	Floor level	Baseline price (0.4463)	Under 150 kW·h	86.09 kW·h
	Middle level	Baseline price + 0.02	151–400 kW·h	
	Top level	Baseline price + 0.12	Over 400 kW·h	
Sichuan Province in 2006	First level	Baseline price	Under 60 kW·h	63.9 kW·h
	Second level	Baseline price + 0.08	61–100 kW·h	
	Third level	Baseline price + 0.11	101–150 kW·h	
	Fourth level	Baseline price + 0.16	Over 150 kW·h	
	Note: Due to the lack of data, average consumption is calculated using data on the whole country.			
The remaining provinces in China in 2012	Free level	Free	Under 15 kW·h	Families on social assistance
	Floor level	Baseline price	Under 240 kW·h	80%
	Middle level	Baseline price + 0.05	241–400 kW·h	90%
	Top level	Baseline price + 0.3	Over 400 kW·h	-
	Note: Consumption varies among the provinces and the data used in the table are for Beijing. The first breakpoints of most provinces fall between 150 and 200 kW·h per month/family, while the second fall between 350 and 400 kW·h per month/family.			

**Table 2 ijerph-18-02620-t002:** The induced effect of IBP on green innovation.

Variables	All Technological Patents		Energy-Efficient Patents	
	(1)	(2)	(3)	(4)
IBP	0.456 **	0.294 ***	0.799 ***	0.576 ***
	(2.17)	(2.75)	(2.97)	(3.72)
Level of economy		0.843 ***		0.856 ***
		(8.68)		(7.26)
Economic structure		0.339 ***		0.409 ***
		(5.13)		(4.64)
Economic environment		−0.278		−1.079 ***
		(–1.03)		(−2.85)
International trade		−2.108 **		−4.777 ***
		(–2.19)		(–3.64)
Human resources		−0.00501		0.0182
		(−0.35)		(0.89)
Investment in R&D		0.428 ***		0.449 ***
		(7.30)		(6.75)
Constant	7.300 ***	−2.588 ***	4.363 ***	−5.728 ***
	(30.10)	(−5.37)	(18.45)	(−8.72)
Observations	377	377	377	377

Note: ***, ** denote significance at the 0.01, 0.05, and 0.1 levels, respectively. T-values are in parentheses and *p*-values are in brackets.

**Table 3 ijerph-18-02620-t003:** The biased effect of the IBP.

	(1)	(2)	(3)	(4)
Variable	Proportion of Green Innovation		Utility Models	Inventions
IBP	2.127 ***	1.978 ***	2.289 **	0.852
	(3.01)	(2.79)	(2.33)	(1.22)
Level of economy		0.530	0.265	1.209 **
		(1.20)	(0.53)	(2.51)
Economic structure		0.367	1.130 **	−0.0866
		(0.93)	(2.32)	(−0.23)
Economic environment		−5.908 ***	−5.253 **	−7.437 ***
		(−3.44)	(−2.33)	(−4.32)
International trade		−15.24 **	−18.75 **	−17.20 ***
		(−2.50)	(−2.45)	(−2.92)
Human resources		0.161 *	0.0122	0.351 ***
		(1.73)	(0.10)	(3.86)
Investment in R&D		0.337	–0.189	−0.394
		(1.18)	(−0.58)	(−1.44)
Constant	5.587 ***	4.394	10.73 ***	−2.549
	(24.00)	(1.62)	(3.54)	(−0.79)
Observations	377	377		

Note: ***, **, * denote significance at the 0.01, 0.05, and 0.1 levels, respectively. T-values are in parentheses and *p*-values are in brackets.

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
