# Peer review of "Induced Effect of Environmental Regulation on Green Innovation: Evidence from the Increasing-Block Pricing Scheme"

_ijerph, 2021, doi:10.3390/ijerph18052620_

Round 1

Reviewer 1 Report

The paper has improved for sure. I am glad to see that the authors are thinking more carefully about what they are doing. However, now in the new version, the pre-trend graphs and tables have disappeared. More fundamentally, there remains this issue that they are trying to measure the treatment effect of a program at the level of entire provinces. I think this is a dodgy exercise because many things can change at the province level from one year to another. The authors completely fail to see that, and pre-trends don't do anything to tackle this problem. 

Author Response

We have added the pre-trend graphs and tables to the supplementary files as Figure S2 and Table S3.

It is a good practice to address this issue from a more micro perspective. However, there are some problems before we realized it in a more micro-level, such as prefecture or county level:

  1. The IBP scheme was experimented on a provincial basis, and official patent data are also collected on a province basis. For now, no official patent data is offered in a city level, so city level data used by existing literature are manually searched, which will lead to significant bias in results. Consequently, it is acceptable to conduct the analysis the impact of IBP on green technological innovation on a province level.
  2. It is well known that there is geographical convenience regarding patent applications in China that most capital in each province offer some beneficial policies for enterprises or individuals applying for patent locally which results in a significant concentration of patents in capitals. If city data were used for empirical analysis, conclusions may be misleading.

For all the above reasons , we are unable to discuss this topic on a more micro level for now. And we will keep an eye on this issue and write a paper about it as long as city level data is available. 

Reviewer 2 Report

This is an interesting study and its quality is good. However, I would like to see the following improvements before making a concrete decision.

1, The authors should use the abbreviations appropriately, see for example IBP in the abstract.

2. The authors should avoid using multiple references in a sentence. To this extent, no more than 3 references in a sentence are appropriate.

3. The authors should provide a footnote that they are willing to share their raw data set in Excel with those who wish to replicate the results of this research.

4. The authors use DiD approach this approach causes loss of information in the data but it seems that the authors are not concerned about this issue.

5. The authors should provide a separate data section that should explain the unit measurements of variables and their sources.

6. The policy recommendations are rather broad, they should be refined,

Author Response

This is an interesting study and its quality is good. However, I would like to see the following improvements before making a concrete decision.

1: The authors should use the abbreviations appropriately, see for example IBP in the abstract.

A: We added explanations when the abbreviations appeared for the first time.

2.The authors should avoid using multiple references in a sentence. To this extent, no more than 3 references in a sentence are appropriate.

A: We have adjusted the references and no more than 3 references in a single sentence.

3.The authors should provide a footnote that they are willing to share their raw data set in Excel with those who wish to replicate the results of this research.

Q: We have added such a footnote in line195 that we provided all data in the excel files for replications.

4.The authors use DiD approach this approach causes loss of information in the data but it seems that the authors are not concerned about this issue.

A: Regarding information loss, we used simple individual-effect model which causes no information loss as robustness check in the previous version. But for simplicity, we have deleted that part in the later version. Results of induced effect are shown in the tables in the attachmenet. Biased induced effect of energy-efficient patent is shown in column (1) and (2) in Table 2, and that of utility model and invention are shown in column (3) and (4). All results share similar level and significance with those using Did method.

5.The authors should provide a separate data section that should explain the unit measurements of variables and their sources.

A: We have added Table S1 to explain the definition, measurement and sources of all variables.

6.The policy recommendations are rather broad, they should be refined.

A: We have added several refined policy implications in line 465-477.

Round 2

Reviewer 2 Report

Accept as it is

This manuscript is a resubmission of an earlier submission. The following is a list of the peer review reports and author responses from that submission.

Round 1

Reviewer 1 Report

I think that the topic of the paper here presented – Induced Effect of Environmental Regulation on Green Innovation: Evidence from the Increasing - block Pricing Scheme - has some interest for publication in Int. J. Environ. Res. Public Health.

Interesting article. The article deals with a very important and current topic concerning patented innovations, in particular those focused on ecology. All pro-ecological activities are highly desirable due to the rapid degradation of the natural environment and a low level of awareness and social responsibility. That is why innovations, and green innovations in particular, have become an important element in the development of enterprises, the economy and the country.

The article contains interesting considerations and research, indicates dependent variables, factors influencing the evaluation of innovation, etc. Examples of high and low quality innovation are missing. Consider the statement or clarification of applications or more important for the economy, environment, society, etc. is: innovation, high quality and innovation of high quality results achieved?

Author Response

Dear reviewer:

We are really honored to get your comments on our imperfect paper. After reading the comments, we finally realized there is a long way to go if we want to perfect our work. Although we didn’t know you, we will be more professional and rigors towards every forthcoming study just like you. All corrections are highlighted in yellow.

I think that the topic of the paper here presented – Induced Effect of Environmental Regulation on Green Innovation: Evidence from the Increasing - block Pricing Scheme - has some interest for publication in Int. J. Environ. Res. Public Health.

Interesting article. The article deals with a very important and current topic concerning patented innovations, in particular those focused on ecology. All pro-ecological activities are highly desirable due to the rapid degradation of the natural environment and a low level of awareness and social responsibility. That is why innovations, and green innovations in particular, have become an important element in the development of enterprises, the economy and the country.

Q: The article contains interesting considerations and research, indicates dependent variables, factors influencing the evaluation of innovation, etc. Examples of high and low quality innovation are missing. Consider the statement or clarification of applications or more important for the economy, environment, society, etc. is: innovation, high quality and innovation of high quality results achieved?

A: For example, if one technic improves the battery of an electric car and extend the miles from 600km to 1000km, that technic is qualified for inventions (high quality). If a producer makes a model of electric car but still uses the 600km battery, that technic is utility model (low quality). For macro economy, only if we have more high-quality innovations and grasp the essence of technics are we able to fight against technological barrier set by other countries and keep independent in the global economic crisis. For the micro economy, products with more innovative technics sell for higher price and accordingly more added value, which finally benefit the people. For the environment, environmental protection is so urgent needed that only big breakthrough or high-quality innovations can favor the environment and keep human being’s welfare at the same time. For the society, breakthrough or major improvement are more likely to spillover spatially and spread rapidly. Even the children will become more creative by the encouragement of the society.    

Reviewer 2 Report

I applaud this effort to dive into the induced innovation effects of increased electricity prices in China. However, the paper as it currently stands suffers from important issues. 

  • The first main issue relates to the clarity of the writing. The introduction in particular would need a complete rehaul. After reading the introduction, we're still left wondering what the paper actually does (i.e., what's the data, what's the method, and in particular the empirical strategy). We're also left wondering what's the difference between the induced effects and the "biased" effect. It's only much later on that we understand the authors mean, i.e. total number of green patents for the former and the total number of green patents divided by the total number of patents for the latter. Mentioning the marginal rate of substitution and aspects relating to productivity is very misleading and confusing here. This paper is about and only about patents, not about productivity or MRS of different factor inputs in a production function. The introduction also fails to explain what "quality" means here. 
  • I think the argument that inventions are of higher quality than utility patents is not sufficiently backed up and developed. It is simply dropped and the reader is asked to accept it. In fact, what the authors say about an idea being tangible or not, would make me lean towards thinking that utility models are higher quality here... We'd have to agree as well what "quality" is, or rather what it is we think we should capture. If quality is about technologies with concrete impacts on consumers or industrial activities, then tangibles would be seen as higher quality, wouldn't they? I'd be open to being convinced that inventions are of higher quality but I'd need more and better arguments. 
  • The next big issue is regarding the empirical strategy. Why should we believe the effect you're measuring is the causal effect of the policy? I would be much more modest and make sure to think through possible confounders and caveats of your current approach. Here, the unit of observation is province-year. The difficulty is that this is quite a high level of aggregation, i.e. provinces are large entities and a lot of things happen at their level over time. Thus, we must worry about something changing in the treated provinces when the treatment also occurs. Many controls are added but also all of these variables may be endogenous here. 
  • The geographical level of the observations also bears questioning. The policy is indeed implemented at the province level, but innovation processes and R&D activities, in particular, are better understood at the level of a country. I suppose firms do not have R&D centers in every province and the decision to invest more R&D in one technology or another also does not happen at the province level. In my mind, it is therefore quite unclear what we should expect in terms of the induced effects of the block pricing scheme when just applied to one province. A better way to investigate the impact of such policy may be on the adoption of energy-efficient technologies, as opposed to the innovation side (i.e. using patents). 
  • The graphs you show of the pre-trends show there is a lot of noise in your data. I would be curious to see these graphs not with growth on the y-axis but with levels instead. 
  • I would also want to see a summary statistics table that compares the treated provinces with the rest of China along all the covariates you have data for. 

To conclude, I think the topic is really interesting but the paper at this stage lacks clarity and rigor.  

Author Response

Dear reviewer:

We really appreciate your comments on our imperfect even poor writing paper. That’s must a tough work. After reading the comments, we finally realized how many details and logic should be improved. And you made us understand that the devil is in the details. Although we didn’t know you, we will be more professional and rigors towards every forthcoming study just like you. All corrections are highlighted in yellow.

Q1: The first main issue relates to the clarity of the writing. The introduction in particular would need a complete rehaul. After reading the introduction, we're still left wondering what the paper actually does (i.e., what's the data, what's the method, and in particular the empirical strategy). We're also left wondering what's the difference between the induced effects and the "biased" effect. It's only much later on that we understand the authors mean, i.e. total number of green patents for the former and the total number of green patents divided by the total number of patents for the latter. Mentioning the marginal rate of substitution and aspects relating to productivity is very misleading and confusing here. This paper is about and only about patents, not about productivity or MRS of different factor inputs in a production function. The introduction also fails to explain what "quality" means here.

A: We have gone through the whole paper, and made several corrections especially to the introduction part. We have display what we paper actually do (i.e., what's the data, what's the method, and in particular the empirical strategy) in line 83-89. And the explanation the difference between the induced effects and the "biased" effect is put forward in both line 36-38 and 86-87. We have deleted the misleading phrase of productivity and MRS. The meaning of quality is explained in line 45-46.

Q2: I think the argument that inventions are of higher quality than utility patents is not sufficiently backed up and developed. It is simply dropped and the reader is asked to accept it. In fact, what the authors say about an idea being tangible or not, would make me lean towards thinking that utility models are higher quality here... We'd have to agree as well what "quality" is, or rather what it is we think we should capture. If quality is about technologies with concrete impacts on consumers or industrial activities, then tangibles would be seen as higher quality, wouldn't they? I'd be open to being convinced that inventions are of higher quality but I'd need more and better arguments.

A: We have deleted the misleading phrase of tangible or intangible which have nothing to our main concern. The definition of invention and utility model in terms of quality is provided by the Patent Law of the People’s Republic of China and put forward in line 215-219.

Q3: The next big issue is regarding the empirical strategy. Why should we believe the effect you're measuring is the causal effect of the policy? I would be much more modest and make sure to think through possible confounders and caveats of your current approach. Here, the unit of observation is province-year. The difficulty is that this is quite a high level of aggregation, i.e. provinces are large entities and a lot of things happen at their level over time. Thus, we must worry about something changing in the treated provinces when the treatment also occurs. Many controls are added but also all of these variables may be endogenous here.

A: We have adjusted the empirical strategy. First of all, we moved the OLS model to the start in each part to confirm the causal effect of the IBP; then we used the generalized DID to isolate other possible factors and further support our conclusion; Lastly, we employed counterfactual method in the robustness test to avoid omitted variables and bias for biased effect. Correction are made in part line 241-243, 262-264 and 293-294 respectively. And the results of baseline OLS regression are added in Table 2, Table 4, line 361-319 and line 352-354.

Q4: The geographical level of the observations also bears questioning. The policy is indeed implemented at the province level, but innovation processes and R&D activities, in particular, are better understood at the level of a country. I suppose firms do not have R&D centers in every province and the decision to invest more R&D in one technology or another also does not happen at the province level. In my mind, it is therefore quite unclear what we should expect in terms of the induced effects of the block pricing scheme when just applied to one province. A better way to investigate the impact of such policy may be on the adoption of energy-efficient technologies, as opposed to the innovation side (i.e. using patents).

A: In the case that the block pricing scheme just applied to one province, we may expect spatial spillover of green innovations besides the induced effect in the treated province. But such spillover will not impact the conclusion of this paper. Because technology only spillover after research and development activities within the province. And the spilled technology may not influence green innovations so much in the rest provinces since no incentives to innovate are offered, that is, innovation highly depends on production system [1]. The expectation that R&D center in one province (control group) and production is conducted in other province (treated group) are extremely rare for the treated province are mostly developed areas and have plenty researches and universities, which made them the first choice to locate a R&D center.

Q5: The graphs you show of the pre-trends show there is a lot of noise in your data. I would be curious to see these graphs not with growth on the y-axis but with levels instead.

A: The graphs are attached in the word file.

Q6: I would also want to see a summary statistics table that compares the treated provinces with the rest of China along all the covariates you have data for.

A: All covariates we have data for are summarized in the following table.

Covariate

Definition

Obs.

Mean

Std

Min

Max

Level of economy

Logarithm of GDP

45

(405)

9.138

(8.474)

0.601

(1.051)

8.148

(5.476)

10.265

(10.808)

Economic structure

Ratio of secondary industry to service sector

45

(405)

0.805

(0.895)

0.085

(0.414)

0.630

(0.494)

0.977

(3.443)

Economic environment

Proportion of scaled firms at a loss in all firms

45

(405)

0.147

(0.231)

0.668

(0.104)

0.069

(0.296)

0.356

(0.521)

International trade

Ratio of FDI to total GDP

45

(405)

0.028

(0.026)

0.018

(0.023)

0.002

(0.001)

0.079

(0.134)

Human resources

Average education years of citizens over the age of 6

45

(405)

8.204

(8.045)

1.105

(1.116)

6.119

(4.693)

11.009

(12.028)

Investment in R&D

Logarithm of the internal expenditures on R&D

45

(405)

9.201

(8.331)

0.996

(10.51)

6.907

(4.424)

11.037

(11.645)

To conclude, I think the topic is really interesting but the paper at this stage lacks clarity and rigor. 

Reviewer 3 Report

The manuscript, by adopting DID method, analyzes the effects of the IBP scheme on green innovation in China. Although the topic is somewhat interesting, the research should have been more carefully designed.

1. The most critical issue is that from which province they get the patent license is not necessarily same to in which province they've developed the new technology especially within one country. However, the manuscript implicitly assumes that the two locations should be same.

2. Even in the case where those two locations are same, we cannot analyze the causal effect of the IBP scheme on local innovation of green technology. The IBP scheme was implemented in Zhejiang, Fujian and Sichuan provinces, which were not randomly selected.  

Author Response

Dear reviewer:

We really appreciate your comments on our imperfect paper. That’s must a tough work. After reading the comments, we finally realized there is a long way to go if we want to perfect our work. Although we didn’t know you, we will be more professional and rigors towards every forthcoming study just like you. All corrections are highlighted in yellow.

Q1: The most critical issue is that from which province they get the patent license is not necessarily same to in which province they've developed the new technology especially within one country. However, the manuscript implicitly assumes that the two locations should be same.

 A: Innovation highly depends on production system, and in turn services the production system. Firm in the provinces implemented the IBP are more motivated to conduct innovations on energy-efficient technology. Even firms in the rest provinces are inclined to apply energy-efficient patent in the treated provinces where related industrial agglomerate and these patents can be quickly applied and transferred to production. What’s more, to encourage R&D activities, local government offer extra award for those apply for related patents.

Q2: Even in the case where those two locations are same, we cannot analyze the causal effect of the IBP scheme on local innovation of green technology. The IBP scheme was implemented in Zhejiang, Fujian and Sichuan provinces, which were not randomly selected.  

A: It is challenging to admit that Zhejiang, Fujian and Sichuan provinces were randomly selected. There is a critical precondition of generalized DID, that is, the treatment group and control group share a common trend on dependent variables [1]. And the precondition relaxes the restriction of randomization [2][3]. So we followed the procedure of Tanaka(2015) and conducted a common trend test in section 4.1. And results indicate that dependent variables in both groups showed similar pattern during 1998-2014.

Reference

[1]Wing, C., Marier, A., 2014. Effects of occupational regulations on the cost of dental services: evidence from dental insurance claims. Journal of Health Economics, 34, 131-143.

[2] Moser P , Voena A . Compulsory Licensing: Evidence from the Trading with the Enemy Act[J]. American Economic Review, 2012, 102.

[3]Hoynes, H., Page, M., Stevens, A. H., 2011. Can targeted transfers improve birth outcomes?: Evidence from the introduction of the WIC program. Journal of Public Economics, 95(7-8), 813-827. doi:10.1016/j.jpubeco.2010.12.006.

[4]Tanaka S . Environmental regulations on air pollution in China and their impact on infant mortality[J]. Journal of Health Economics, 2015, 42(jul.):90-103.

Round 2

Reviewer 2 Report

The methods, empirical strategy, and causal identification are still unsatisfactorily discussed and thought through. Results are misinterpreted.

  • There are a series of sentences that are problematic because they highlight a lack of understanding of what causal identification requires. For example, "we used basic OLS method to confirm the causal relationship between the IBP and increasing energy-efficient patent."There is nothing about an "OLS" that is inherently causal. 
  • You need to start with clearer explanations of what each of your regression buys you. For a start, you say "OLS" and then "generalized DID method". It is inappropriate to use these terms for what you are doing. OLS is an estimator. It says nothing about whether the interpretation of the estimated parameter is causal or not. The difference between your two equations (1) and (2) seem to be only the year fixed effects. but otherwise, it's really the same thing. One is not more "OLS", and the other is not more "DiD". 
  • Good causal identifications are difficult to get by and, in my view, it is still ok to write a paper that examines the effect of X on Y even though the identification is not perfect. But at the very least, your paper MUST highlight the caveats and limitations of the approach. In particular, other confounders. 
  • The scale of the graphs in Figure 2 is unbelievably high. Growth rates of around 40% are really high year-on-year. 
  • The pre-trend graphs you attached with levels instead of growth are hard to read. Given the nature of the data, the y-axis should be logged. You should also probably label the axis. Only attach figures in pdf. 
  • The graphs, though, clearly show that the treated provinces simply have higher patenting, especially in the later years.
  • First, if you were to replicate your results by focusing on a narrower time period around the treatment (say 5 years), you could more credible attribute your effect to the treatment.
  • Second, the fact that there is just more patenting in all categories (not just in energy efficiency) is clearly consistent with the idea that your "treated" provinces are just provinces with more patenting to the controls in general. And, so you also get more patenting in energy efficiency but that probably has nothing to do with IBP.
  • As said before, your pre-trend graphs are not convincing because they look very noisy. But, even if the pre-trend looked perfect, for identification, you'd still have to argue that the treated and control provinces are exposed to exactly the same conditions apart from the IBP for all the years in your sample after the IBP comes into place. Obviously, this is a very hard assumption to defend.
  • The fact that these provinces have well-developed economies with many universities and research centers does in no way ensure that research related to energy-efficient technologies takes place in every province in response to province-level shocks. This is a caveat of your empirical design that you can't really deal with and so at the minimum, you should recognize it and write down a note about it in the paper.  

Reviewer 3 Report

The revised version has improved sufficiently.